# Lactone Stabilized by Crosslinked Cyclodextrin Metal-Organic Frameworks to Improve Local Bioavailability of Topotecan in Lung Cancer

**DOI:** 10.3390/pharmaceutics15010142

**Published:** 2022-12-31

**Authors:** Ting Xiong, Tao Guo, Yaping He, Zeying Cao, Huipeng Xu, Wenting Wu, Li Wu, Weifeng Zhu, Jiwen Zhang

**Affiliations:** 1Key Laboratory of Modern Preparation of TCM, Jiangxi University of Traditional Chinese Medicine, Ministry of Education, Nanchang 330004, China; 2Center for Drug Delivery Systems, Shanghai Institute of Materia Medica, Chinese Academy of Sciences, Shanghai 201203, China; 3College of Pharmacy, Shenyang Pharmaceutical University, Shenyang 110016, China; 4Department of Pharmacy, The First Affiliated Hospital of Zhengzhou University, Zhengzhou 450052, China; 5University of the Chinese Academy of Sciences, Beijing 100049, China

**Keywords:** topotecan, crosslinked cyclodextrin metal-organic framework, stability, melanoma lung metastasis, local bioavailability

## Abstract

The protection of unstable anticancer molecules and their delivery to lesions are challenging issues in cancer treatment. Topotecan (TPT), a classic cytotoxic drug, is widely used for treating refractory lung cancer. However, the therapeutic effects of TPT are jeopardized by its active lactone form that is intrinsically hydrolyzed in physiological fluids, resulting in low bioavailability. Herein, the TPT-loaded crosslinked cyclodextrin metal-organic framework (TPT@CL-MOF) was engineered to improve the local bioavailability of TPT for the treatment of lung cancer. CL-MOF exhibited the efficient loading (12.3 wt%) of TPT with sustained release characteristics. In particular the formulation offered excellent protection in vitro against hydrolysis and increased the half-life of TPT from approximately 0.93 h to 22.05 h, which can be attributed to the host–guest interaction between cyclodextrin and TPT, as confirmed by molecular docking. The TPT@CL-MOF could effectively kill the cancer cells and inhibit the migration and invasion of B16F10 cells in vitro. Moreover, TPT@CL-MOF was efficiently distributed in the lungs after intravenous administration. In an in vivo study using a B16F10 pulmonary metastatic tumor model, TPT@CL-MOF significantly reduced the number and size of metastatic lung nodules at a reduced low dose by five times, and no noticeable side effects were observed. Therefore, this study provides a possible alternative therapy for the treatment of lung cancer with the camptothecin family drugs or other unstable therapeutically significant molecules.

## 1. Introduction

Lung cancer remains one of the most invasive and dreadful diseases worldwide, and more than 70% of lung cancer deaths are attributed to metastatic lung cancer [1,2]. Conventional chemotherapy is one of the main treatment options for lung cancer [3]. However, low delivery efficacy and non-specificity have led to the unacceptable cytotoxicity of normal healthy tissues, which has severely limited its further application. Consequently, there is an urgent need to improve the local bioavailability of antitumor drugs in lung cancer to mitigate the influences of various impediments [4,5].

To date, a number of particle-based delivery systems have been evaluated for lung cancer therapy, such as microspheres [6], liposomes [7], solid lipid nanoparticles [8], polymer nanoparticles [9], mesoporous silica nanoparticles [10], and graphene quantum dots [11]. Metal-organic frameworks are among the different nanoplatforms being employed in drug delivery systems. They have attracted considerable interest in recent years as cancer therapeutic platforms [12,13]. In particular, cyclodextrin metal-organic frameworks (CD-MOFs) have been considered potential vectors because of their superior properties, such as porosity, tunable size, biocompatibility, and functional diversity [14,15]. Previous studies have demonstrated that CD-MOF can be employed to enhance the solubility [16,17,18], stability [19,20,21,22,23,24], and bioavailability [25,26] of numerous drugs. Therefore, it can be used for delivering various types of therapeutic agents. Recently, crosslinked CD-MOF (CL-MOF) nanoparticles were found to be primarily distributed in the lung following intravenous injection, delivering doxorubicin as well as low molecular weight heparin to treat lung cancer [27]. Therefore, CL-MOF can be utilized as a promising pulmonary delivery system. 

Topotecan (TPT), a synthetic analogue of the camptothecin family, has been extensively used in the treatment of lung cancer [28,29]. However, the poor stability of TPT results in its low bioavailability as rapid hydrolysis converts the metastable lactone ring into an inactive carboxylated open ring [30]. Although the conversion of the carboxylate form is a chemically reversible reaction, it mainly results in unidirectional inactivation in physiological environments. To date, research works on the protection of TPT against hydrolysis are mostly focused on lipid-based delivery systems. However, the superior stability of TPT via lipid-based strategies fails to achieve significant efficiency. For example, the “fluid” liposome only remained a 25.97% lactone form of TPT after 48 h of incubation in phosphate-buffered saline (PBS, pH 7.4) [31]. The content of the lactone ring on lipid nanoparticles in an artificial intestinal medium (pH 6.5) seemed to be more than that of free TPT at each predetermined time point, and by contrast, the result was not significant enough [32]. In addition, the application of other sorts of nanocarriers, including mesoporous silica [33] and cyclodextrin complexes [34], to sequester TPT is limited by suboptimal loading, high leakage, and non-specificity [35]. In this regard, the porous channels of CL-MOF play a vital role in protecting the encapsulated TPT in the form of a stable lactone via the host–guest interaction. 

In this study, we developed an alternative pulmonary delivery platform that used CL-MOF to capture and stabilize the physiologically unstable TPT in order to increase its local bioavailability. Subsequently, the TPT-loaded crosslinked cyclodextrin metal-organic framework (TPT@CL-MOF) was systematically characterized, and the release profile and stability tests were assessed under physiological conditions. As a proof-of-concept, we investigated the in vitro inhibitory effects assay of TPT@CL-MOF on murine melanoma cells (B16F10), as well as its therapeutic efficacy in B16F10 metastatic tumor-bearing mice.

## 2. Materials and Methods

### 2.1. Materials

TPT was provided by Dalian Meilun Biotech Co., Ltd. (Dalian, China). γ-cyclodextrin was purchased from MaxDragon Biochem Ltd. (Guangzhou, China). Diphenyl carbonate (DPC) was supplied by Aladdin Reagent Co., Ltd. (Shanghai, China). N, N-dimethylformamide (DMF, SuperDry) was provided by Budweiser Technology Co., Ltd. (Shanghai, China). Potassium hydroxide, polyethylene glycol 20,000, methanol, ethanol, and triethylamine (TEA) was of an analytical grade and obtained from Sinopharm Chemical Reagent Co., Ltd. (Shanghai, China). Cyanine 5 NHS ester (Cy5) was purchased from Xi’an Ruixi Biological Technology Co., Ltd. (Xi’an, Shanxi, China). Fetal bovine serum (FBS) was provided by Gemini (GeminiBio, West Sacramento, CA, USA). Cellulose dialysis membranes (MWCO: 3500 Da) were produced by Shanghai Yuanye Bio-Technology Co., Ltd. (Shanghai, China). Water was purified using a Milli-Q water system (Millipore, Billerica, MA, USA). All reagents used in the experiment were obtained from commercial sources and used without further treatment.

### 2.2. Cell Culture and Animals

B16F10 cells were obtained from the Shanghai Cell Bank, Chinese Academy of Sciences (Shanghai, China). The cells were cultured in Dulbecco’s modified Eagle’s medium (DMEM) containing 10% FBS at 37 °C in an environment of 5% CO_2_. 

Male C57BL/6 mice (six weeks old, 18–22 g, IACUC Application No. 2020-05-ZJW-28) were obtained from the Shanghai Laboratory Animal Center, Chinese Academy of Sciences. All mice were acclimatized for 1 week before the experiments and provided with free access to water and food in a light and humidity-controlled environment. The animal experimental and surgical procedures were performed according to the experimental guidelines issued by the Institutional Animal Care and Use Committee of the Shanghai Institute of Materia Medica, Chinese Academy of Sciences.

### 2.3. Synthesis of CL-MOF

Nanosized CD-MOFs were synthesized following our previously reported methodology [36]. Briefly, in a closed reaction vessel, 3.1 g of CD-MOF and 3.0 g of DPC were incubated at a 1:6 molar ratio in 40 mL of DMF at a speed of 400 rpm for 10 min by a magnetic stirrer (RCT Basic, IKA^®^, Staufen, Germany). Next, 450 μL of TEA was added as a catalyst to the above solution to speed up the reaction. Then, the mixture proceeded for 24 h under stirring at 80 °C. On the completion of the reaction, the mixture was cooled down to room temperature, and subsequently, 95% of ethanol was added to terminate the reaction. Then, CL-MOF was washed twice with 50% of ethanol, distilled water, and acetone for purification and then dried overnight at 40 °C under a vacuum.

### 2.4. Drug Loading

A total of 30 mg of TPT was dissolved in 3 mL of distilled water, followed by the addition of CL-MOF powders (50 mg). The mixture was incubated under magnetic agitation at a speed of 300 rpm for 12 h at an ambient temperature. After that, the products were collected by centrifugation and washed with water to remove the unloaded TPT. Finally, the TPT-loaded crosslinked cyclodextrin metal-organic framework (TPT@CL-MOF) was obtained after lyophilization. To determine the TPT loading content, the absorbance values of the nanoparticles at 422 nm in a 0.1 M NaOH aqueous solution were tested using a UV-Vis spectrometer (UH5300, Japan). The loaded mass of TPT was obtained against a calibration curve (Appendix A), and the drug payload was calculated using Equation (1).
(1)$$\mathrm{Payload}\;\left(\%\right)=\frac{\mathrm{Mass}\;\mathrm{of}\;\mathrm{TPT}\;\mathrm{loaded}\;\mathrm{in}\;\mathrm{CL}\text{-}\mathrm{MOF}}{\mathrm{Mass}\;\mathrm{of}\;\mathrm{TPT}@\mathrm{CL}\text{-}\mathrm{MOF}}\times 100\;$$

### 2.5. Characterizations of TPT@CL-MOF

#### 2.5.1. Particle Size and Zeta Potential Measurement

Both the size distribution and zeta potential of the samples in the water were explored in triplicate by the dynamic light scattering (DLS) method (Zetasizer Nano ZS90, Malvern, UK) at 25 °C.

#### 2.5.2. Scanning Electron Microscopy (SEM)

The surface morphology of CL-MOF and TPT@CL-MOF was characterized through SEM (FlexSEM1000, Hitachi, Japan). The samples were covered with a thin layer of gold and observed at an appropriate magnification.

#### 2.5.3. Thermogravimetric Analysis (TGA)

The thermal degradation of the samples was carried out via the TGA method (Pyris 1, PerkinElmer, Waltham, MA, USA) in a temperature range from 25 to 600 °C with a 20 °C/min increasing rate under a nitrogen gas atmosphere (20 mL/min).

#### 2.5.4. Differential Scanning Calorimetry Analysis (DSC)

The phase transition of TPT was determined by DSC analysis (Q2000, TA Instruments, New Castle, DE, USA). The powders, including TPT, CL-MOF, and TPT@CL-MOF, were sealed in aluminum discs and measured in the temperature range from 20 to 400 °C with a heating rate of 10 °C/min.

#### 2.5.5. Fourier Transform Infrared Spectroscopy (FTIR)

FTIR spectra of the samples were collected by a spectrometer (Nicolet FTIR 6700, Thermo Fisher Scientific, Waltham, MA, USA). The samples and potassium bromide were mixed in a ratio of about 1:10 and compressed into a tablet. Each sample was recorded at a resolution of 4 cm^−1^ in a wavenumber range from 4000 to 500 cm^−1^.

### 2.6. In Vitro Release 

The release behaviors of TPT@CL-MOF were performed via the dialysis technique under physiological conditions at pH 7.4. Briefly, TPT@CL-MOF (5 mg) and TPT were sealed in dialysis bags, respectively. Each sample was soaked in PBS (pH 7.4, 40 mL) under continuous shaking (100 rpm) at 37 °C. Aliquots (1 mL) of the solution from the release medium were withdrawn at different time points (0.25, 0.5, 1, 2, 4, 6, 8, 12, 24, 36, and 48 h), while an equal amount of fresh buffer was simultaneously substituted. The content of the cumulatively released TPT was measured using a Luna C18 (4.6 mm × 250 mm, 5 μm) column at 30 °C. Acetonitrile and 0.1% trifluoroacetic acid aqueous solution (20:80, *v*/*v*) was used as the mobile phase with a flow rate of 0.8 mL/min and injection volume of 10 μL. The detection wavelength was 228 nm. 

### 2.7. In Vitro Stability

The hydrolysis kinetics of TPT was evaluated to investigate the protective effect of CL-MOF. TPT or TPT@CL-MOF was incubated in PBS (pH 7.4) at a concentration of 0.5 mg/mL. At defined time intervals (0.5, 1, 2, 4, 6, 8, 12, and 24 h), samples were taken, and the lactone ring content of TPT was analyzed by high-performance liquid chromatography (1260, Agilent Technologies, Santa Clara, CA, USA) according to a previous report [32]. 

### 2.8. Molecular Docking

AutoDock Vina 1.1.2 was applied to perform the molecular docking study [37]. The three-dimensional structure of TPT was acquired from the chemistry database of PubChem (CID: 60700), and the structure of CL-MOF was constructed manually by connecting -OHs with -C=O- of the CD-MOF model extracted from the reported single crystal structure [38]. First, an energy minimization protocol was employed to prepare the TPT model. Then, the TPT molecule was docked to the CL-MOF model in a rigid docking way. In this protocol, the Lamarckian genetic algorithm, in line with a grid-based energy assessment, was utilized to pre-calculate the grid maps in terms of the interatomic potentials of all-atom models. A grid map of dimensions 20 Å × 20 Å × 20 Å with the active position (0.291, 24.253, 20.748) was settled, and in the meantime, other parameters were set as a default. The optimized docking procedure with relatively low energies was used for subsequent analysis.

### 2.9. Cytotoxicity Assay

The cytotoxicity of TPT and TPT@CL-MOF was assessed against B16F10 cells by a WST-8 tetrazolium salt (CCK-8) assay. Briefly, B16F10 cells (2 × 10^4^ cells per well) were cultured in 96-well plates for 24 h. Afterward, the old medium was discarded, and the cells were exposed to different concentrations of TPT@CL-MOF (0.81–406 μg/mL) and TPT (0.1–50 μg/mL) for 24 h. Then, a CCK-8 solution was added to each well and sequentially incubated for 1.5 h, and the absorbance was detected at 450 nm using a microplate reader (Multiskan GO, Thermo Fisher Scientific, USA). Six parallel wells were tested for each sample. The B16F10 cells without samples were used as control, and the background absorbance of the medium was represented as a blank group. The cell viability (%) was calculated using the following Equation (2).
(2)$$\mathrm{Cell}\;\mathrm{viability}\;\left(\%\right)=\frac{A_{\mathrm{sample}}-A_{\mathrm{blank}}}{A_{\mathrm{control}}-A_{\mathrm{blank}}}\;\times \;100\;$$

### 2.10. Cell Migration and Invasion Assay

To perform a wound healing study, 2.5 mL of the B16F10 cells at a density of 4 × 10^4^ cells per well were added to a six-well plate and incubated for 24 h. Cell layers near the confluence were scratched in a straight line by a sterile pipette tip, and cells were gently washed three times with PBS. Then, the cells were incubated with serum-free fresh medium containing TPT and TPT@CL-MOF at the TPT concentration of 1 μg/mL for 24 h, respectively. The images of the wound healing distance were collected using an inverted optical microscope (Leica DMi1, Wetzlar, Germany), and the wound closure rate was calculated according to the following Equation (3):(3)$$\mathrm{Wound}\;\mathrm{closure}\;\mathrm{rate}\;\left(\%\right)=\frac{{\mathrm{Distance}}_{\mathrm{at}\;24\;h}}{{\mathrm{Distance}}_{\mathrm{at}\;0\;h}}\;\times \;100\;$$

To assess the anti-invasion ability, 100 μL of B16F10 cells suspension (1 × 10^5^ cells) in serum-free medium was added to the top chamber of the Transwell insert (Corning, Corning, NY, USA) coated with Matrigel (BD, Bioscience, Rockville, MD, USA), and was gently placed to the lower chamber of 750 μL DMEM containing 20% FBS. After starvation culture for 2 h, aliquots of 100 μL serum-free medium containing TPT or TPT@CL-MOF (at an equivalent TPT dose of 1 μg/mL) were added to treat the cells for 24 h. The cells that failed to invade the upper chambers were swept by a cotton swab, and then the chamber was dipped in 4% paraformaldehyde to fix the penetrated cells on the bottom of the membrane of the chamber and stained with 0.5% crystal violet for 20 min. Finally, the images were acquired by an inverted optical microscope (Leica DMi1, Wetzlar, Germany), and the crystal violet was eluted with 33% acetic acid and quantified at 570 nm by Varioskan Flash Multimode Reader (Thermo Fisher Scientific, USA). The relative invasion rate was calculated in accordance with the following Equation (4): (4)$$\mathrm{Relative}\;\mathrm{invasion}\;\mathrm{rate}\;\left(\%\right)=\frac{{A\;}_{\mathrm{sample}}}{A_{\;\mathrm{control}}}\;\times \;100\;$$

### 2.11. Biodistribution Study

To explore the in vivo distribution of TPT@CL-MOF, the Cy5-labeled TPT@CL-MOF was prepared according to our previous report [27]. In brief, 100 μL of B16F10 cells suspension (2 × 10^5^ cells) were intravenously injected into C57BL/6 mice to establish a metastatic melanoma lung model. On day 20, following the intravenous injection of B16F10 cells, the mice were injected with normal saline and Cy5-labeled TPT@CL-MOF via the tail vein. After administration, the mice were sacrificed at 0, 1, 2, and 4 h. Subsequently, the lungs, livers, spleens, hearts, and kidneys were collected. The semi-quantitative assessment of the fluorescence intensity of the tissues was achieved using an in vivo imaging system (PerkinElmer, USA).

### 2.12. Anti-Metastatic Assays

To assess the anti-metastasis effect of TPT@CL-MOF, a pulmonary melanoma metastasis model was established, as described above. After 3 days, the mice were randomly divided into four groups and administered intravenously with saline, free TPT (5 mg/kg), or TPT@CL-MOF (with TPT at 2.5 or 1 mg/kg) via the tail vein. The administration frequency was once every three days for five sequential cycles. Meanwhile, the mice’s body weights were monitored for 20 days. After treatment for 21 days, the mice were sacrificed. Lung tissues were weighed and photographed. In addition, the metastatic lung foci were counted. The harvested organs (lungs, livers, spleens, hearts, and kidneys) were embedded in a 10% formalin solution for hematoxylin and eosin (H&E) staining. 

### 2.13. Statistical Analysis

All experimental data were shown as means ± standard deviations. Statistical analysis was executed with one-way ANOVA followed by Tukey’s test for multiple groups. *p*-values less than 0.05 were deemed to be a statistically significant difference.

## 3. Results and Discussion

### 3.1. Preparation and Characterizations of TPT@CL-MOF

CL-MOF efficiently encapsulated TPT with a payload of 12.3%, which can be attributed to the superior properties that CL-MOF offers in terms of adsorption. The successful loading of TPT was confirmed via the following experiments.

As shown in SEM (Figure 1A) and DLS (Figure 1B), the synthesized CL-MOF possessed a typical cubic structure with an average size diameter of 244.9 ± 19.2 nm and zeta potential of about −35.9 ± 0.4 mV, while TPT@CL-MOF also displayed a cubic shape with an average size of about 253.3 ± 18.7 nm, and zeta potential of −22.8 ± 1.1 mV. Therefore, the loading process of TPT did not significantly affect the morphology and particle size of CL-MOF. Furthermore, TPT-loading slightly changed the zeta potential of particles, suggesting that TPT was mainly encapsulated into the pore of CL-MOF.

As displayed in Figure 1C, approximately 10% weight loss of CL-MOF was observed from 30 to 200 °C, indicating the loss of water molecules and solvation. The decomposition pattern of TPT@CL-MOF was similar to that of CL-MOF at 200–400 °C. In addition, the gap of weight loss consisting of the drug load between TPT@CL-MOF and CL-MOF was about 15%, which originated from the intrinsic thermal decomposition of CL-MOF and indicated the presence of the drug in its pores. 

DSC images were acquired to explore the physical state of TPT in TPT@CL-MOF, using TPT and empty CL-MOF as a reference (Figure 1D). The raw TPT displayed a peak at approximately 230 °C, which was consistent with its melting point and corresponded to its crystalline nature. On the contrary, this peak was not observed in both CL-MOF and TPT@CL-MOF at the same temperature range. The endothermic peak of TPT in TPT@CL-MOF completely disappeared, indicating that TPT molecules are present inside the CL-MOF in an amorphous state.

The FTIR spectrum of TPT@CL-MOF produced several new characteristic bands for TPT compared with the spectrum of CL-MOF, which can be attributed to the stretching vibration of C-N (1594 cm^−1^) and skeleton vibration of the benzene ring (1656 cm^−1^) from the distinctive characteristic spectral bands of pure TPT, suggesting the presence of TPT (Figure 1E). 

The above tests show that TPT was successfully loaded into CL-MOF particles with high loading.

### 3.2. In Vitro Drug Release and Chemical Stability Test

TPT is a promising candidate for a sustained-release formulation due to its unique S-phase-specific release pattern and extreme instability under physiological conditions. TPT could be chemically stabilized, and its release could be prolonged using a sustained-release formulation. As depicted in Figure 2A, the cumulative release amount of free TPT rapidly reached over 80% within 4 h at pH 7.4. Importantly, it was found that TPT@CL-MOF showed a relatively sustained-release pattern, and the releasing ratio was 80.46% for 48 h. For the first-order kinetics, Higuchi models and Ritger–Peppas models were employed to simulate the release of TPT. The released data of TPT@CL-MOF were in line with the first-order kinetic model (Appendix A). The main mechanism could be attributed to the diffusion of the cargo from the CL-MOF matrix. Hence, the porous structures of CL-MOF can efficiently restrict the undesirable leakage of TPT under physiological conditions, minimizing the side effects of chemotherapeutic drugs and rendering it an ideal drug delivery system.

Embedding sensitive drugs into cavities is a widely accepted method of preventing degradation [20,39,40]. The lactone ring structure is a vital active form of TPT for inhibiting topoisomerase I activity [41]. Nevertheless, it is unstable and could easily hydrolyze to carboxylate under a neutral blood fluid after being administrated intravenously. As illustrated in Figure 2B, the lactone fraction between the TPT solution and TPT@CL-MOF was measured at 37 °C in PBS. Degradation data were fitted on biexponential decay equations, and hydrolysis half-lives (t_1/2_) were calculated in accordance with the equations (Appendix A). The hydrolysis of free TPT generated rapidly at pH 7.4 with a short t_1/2_ of about 0.93 h. Fortunately, more lactone fractions of TPT were supervised from TPT@CL-MOF than free TPT at every designated time point in the PBS (pH 7.4), which possessed a long t_1/2_ of 22.05 h. In general, the lactone-active form of TPT can be easily converted into an inert carboxylate structure in a neutral release medium. Compared with free TPT, the longer t_1/2_ of TPT@CL-MOF was attributed to the protection of the shell of CL-MOF, reducing the release of TPT into the physiological medium and decreasing the hydrolytic degradation. However, a portion of TPT still existed in the opening-ring carboxylate form due to the released TPT from TPT@CL-MOF, even if they were initially encapsulated in the cavity. 

Here, we found the confinement effect of CL-MOF on the degradation of TPT in physiological conditions. The effects of drug restriction attributed to the high affinity between the pockets of the CD could play a key role in improving stability [34,42]. Hence, molecular docking was applied to further study the protection mechanism of TPT in CL-MOF. It can be seen from the docking results that TPT was trapped inside the CD molecular pair of CL-MOF with a higher probability (Figure 2C), and the docking free energy was −8.9 kcal/mol, thus, presented an attractive affinity resulting from the host–guest interaction between CD and TPT. The open ring structure of TPT effectively covered by the CD pair reduced the probability of external water molecules intruding. Additionally, it also lessened the possibility of drug escape and avoided degradation in the physiological environment.

In vitro extended-release and stability results showed that CL-MOF extended the release time and increased the stability of TPT in the physiological environment compared with the free drug, which greatly reduced the exposure of the inactive opening-ring carboxylate of TPT, thus providing the foundation for the subsequent stages, which resulted in increased anticancer activity in vivo.

### 3.3. In Vitro Inhibitory Effects of TPT@CL-MOF

Biocompatibility is an indispensable concern for an ideal drug vector in biomedical applications. Therefore, it is necessary to test the cytotoxicity of CL-MOF in B16F10 cells. As expected, after incubation for 24 h with nanoparticles, CL-MOF exhibited no obvious cellular toxicity, even though the concentration of nanoparticles reached 500 μg/mL (Figure 3A), indicating good cytocompatibility. In our previous study, CL-MOF showed an exceedingly low hemolysis ratio, even though the concentration reached up to 1000 μg/mL, suggesting their admirable blood compatibility as a safe delivery vehicle [36]. It demonstrated low cytotoxicity and good hemocompatibility, which is mandatory for intravenous administration.

The cell viability of TPT@CL-MOF was systematically investigated on B16F10 cells to verify the potential antitumor effect. TPT-based formulations all depicted obvious dose-dependent cytotoxicity (Figure 3B). The half maximal inhibitory concentration (IC_50_) of TPT@CL-MOF was 4.664 μg/mL, which was slightly lower than that of TPT (4.961 μg/mL) (Appendix A). Therefore, TPT@CL-MOF suggested a better antitumor effect than free TPT against B16F10 cells in vitro due to the effective immobilizing and protecting of TPT. It is worth noting that the cytotoxicity of TPT@CL-MOF at a dose of 50 μg/mL was slightly inferior to that of free TPT, which could be explained by decreasing the release rates of TPT by CL-MOF. 

To assess the anti-migration effect of TPT@CL-MOF, wound healing assays were conducted. In Figure 3C, the migration distances of the scratch at the original time point and after 24 h of treatment were recorded. The wound closure rates of blank, TPT, and TPT@CL-MOF were 67.40 ± 4.59%, 28.87 ± 3.40%, and 17.66 ± 4.31%, respectively (Figure 3E). Obviously, the TPT@CL-MOF group revealed the most significant inhibition effect on B16F10 cell motility. Similarly, the Transwell assay was performed to evaluate the invasion inhibition, as shown in Figure 3D. A mass of B16F10 cells rapidly invaded the lower chamber in the control group, implying that B16F10 cells were highly invasive and aggressive. Importantly, the quantitative analysis of TPT and TPT@CL-MOF treatments displayed that the number of invading cells significantly decreased with a relative invasion rate of 55.57 ± 2.67% and 48.97 ± 1.77%, respectively (Figure 3F), which may be due to the inhibitory activity of TPT on the cells.

### 3.4. Lung Distribution of Nanoparticles

To investigate the in vivo distribution of nanoparticles after intravenous injections in a mice melanoma metastasis model, a near-infrared fluorescent dye (Cy5) was linked to TPT@CL-MOF for ex vivo imaging. Cy5 linked TPT@CL-MOF rendered a higher fluorescence intensity in the lungs after 2 h of injection (Figure 4A), and semi-quantitative results also suggested that the fluorescence intensity of nanoparticles in the lungs was stronger than other organs (Figure 4B), indicating the successful drug delivery to the lung by the nanocarriers. Additionally, TPT@CL-MOF was also projected to be more highly accumulated in the lung after 1 h and gradually decreased after 4 h, possibly due to the metabolism of the nanoparticles (Appendix A).

### 3.5. In Vivo Antitumor Efficacy of TPT@CL-MOF in Metastatic B16F10 Tumor-Bearing Mice

Due to the excellent stability improvement of TPT using CL-MOF, we conducted specific therapeutic studies on lung metastasis in vivo. After 3 days of B16F10 melanoma cell implantation in the mice, all preparations were dosed in specified treatment regimens, as shown in Figure 5A. The variation in body weight was considered a helpful parameter to estimate the in vivo toxicity of the nanoparticles. No significant body weight reduction was detected in the TPT@CL-MOF 1 and TPT@CL-MOF 2.5 group compared with the TPT group, suggesting the minimal toxicity induced by TPT@CL-MOF (Figure 5B). Fortunately, mice treated with TPT@CL-MOF (TPT at 2.5 or 1mg/kg) experienced significantly fewer tumor metastasis (Figure 5C) and lighter lung weight (Appendix A). In sharp contrast, TPT@CL-MOF 2.5 (TPT at 2.5 mg/kg) presented a superior inhibition of suppressing lung metastasis at a relatively low dose of TPT among all the groups. The TPT@CL-MOF 1 (TPT at 1 mg/kg) mediated the same lung metastasis area compared to TPT 5 (TPT at 5 mg/kg), which reduced the effective dose by five times (Figure 5D). In conclusion, CL-MOF is a promising carrier to reduce the dose and side effects of chemotherapeutic drugs.

H&E staining revealed that the saline group contained extensive tissue necrosis in the lung, whereas TPT@CL-MOF groups (TPT at 2.5 or 1 mg/kg) exhibited no obvious vigorous tumor growth after treatment (Figure 6). Except for the lung, no obvious damage indications and pathological features were detected in other main organs, including hearts, livers, spleens, and kidneys in each group, indicating the high histocompatibility and good biocompatibility of TPT@CL-MOF.

## 4. Conclusions

In summary, this study highlighted the effectiveness of the encapsulation of TPT within the porous structure of CL-MOF to enhance its local bioavailability for treating lung cancer. It was found that TPT@CL-MOF not only inhibited the restrained burst drug leakage but also effectively improved the stability of TPT in physiological conditions by increasing the half-life from approximately 0.93 h to 22.05 h. In addition, TPT@CL-MOF exhibited excellent anticancer effects. TPT@CL-MOF significantly inhibited the migration and invasion of B16F10 cells in vitro and suppressed tumor growth with equivalent efficacy at a 5-fold reduced dose on a B16F10 pulmonary metastatic tumor model. Importantly, TPT@CL-MOF possessed excellent biocompatibility in recipient mice. Overall, TPT@CL-MOF might act as a unique and promising nano platform for pulmonary metastatic cancer treatment.

## Figures and Tables

**Figure 1 pharmaceutics-15-00142-f001:**
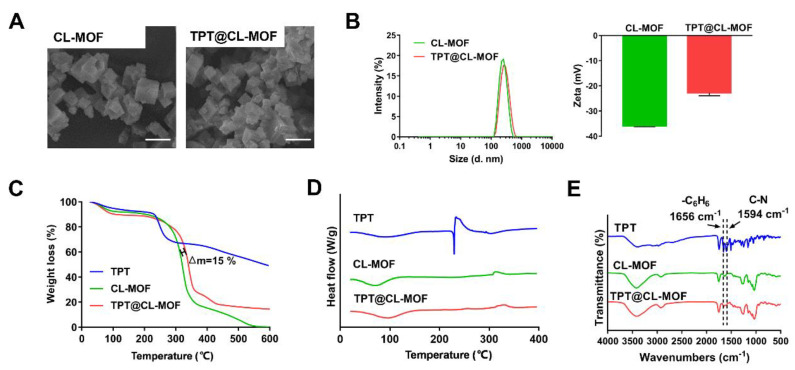
Characterizations of TPT@CL-MOF. SEM images (**A**) of CL-MOF and TPT@CL-MOF. Scale bars, 500 nm. Size diameter and zeta potential (**B**) of CL-MOF and TPT@CL-MOF. TGA curves (**C**), DSC curves (**D**), and FTIR spectra (**E**) of TPT, CL-MOF, and TPT@CL-MOF.

**Figure 2 pharmaceutics-15-00142-f002:**
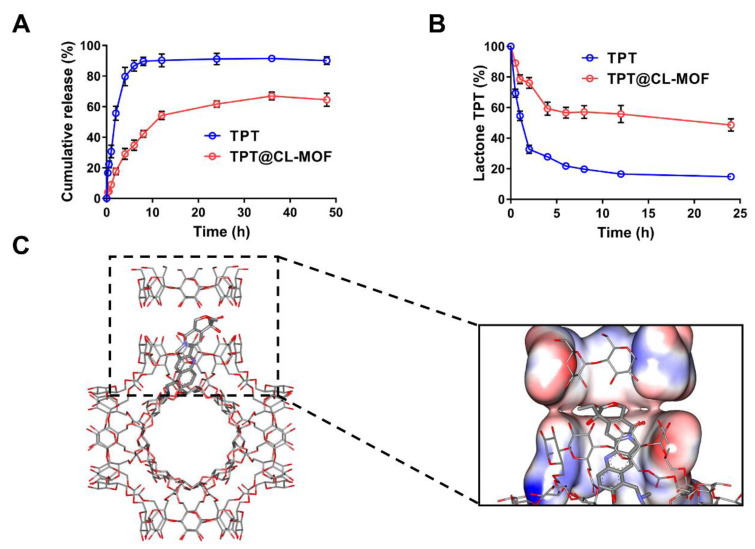
CL-MOF was effective as a drug carrier for TPT’s slow release and stability protection. Cumulative release profiles of TPT (**A**) and the protection effect of TPT@CL-MOF (**B**) in the Buffers with pH 7.4 (*n* = 3). Conformation of TPT molecule (rod model) distributed in CL-MOF (**C**).

**Figure 3 pharmaceutics-15-00142-f003:**
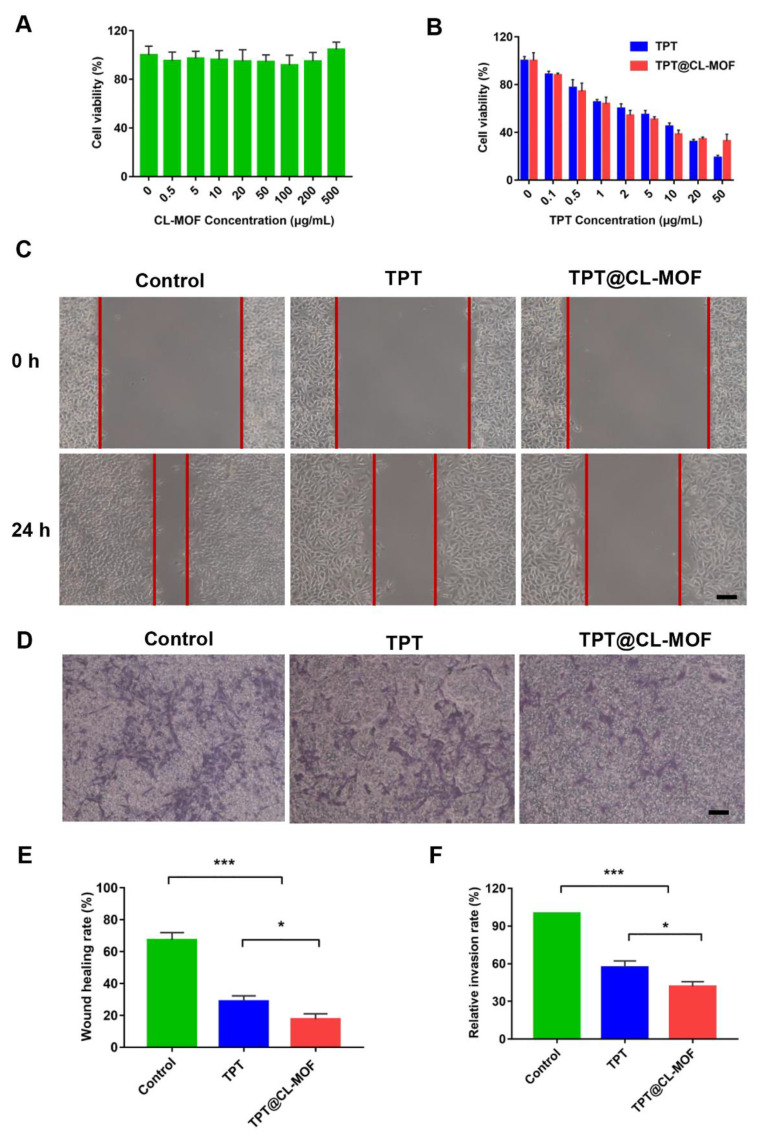
In vitro inhibitory effects of TPT and TPT@CL-MOF on the B16F10 cells. (**A**) The cytotoxicity of CL-MOF in melanoma cells (*n* = 6). (**B**) Cell viability of TPT and TPT@CL-MOF treated B16F10 melanoma cells (*n* = 6). Images (**C**) and Wound healing rates (**E**) of B16F10 melanoma cells (*n* = 3). The distance between the two red lines in (**C**) is the wound healing distance. Invasion images (**D**) and Invasion rates (**F**) of B16F10 melanoma cells (*n* = 3). Scale bars, 100 μm. * *p* < 0.05, *** *p* < 0.001.

**Figure 4 pharmaceutics-15-00142-f004:**
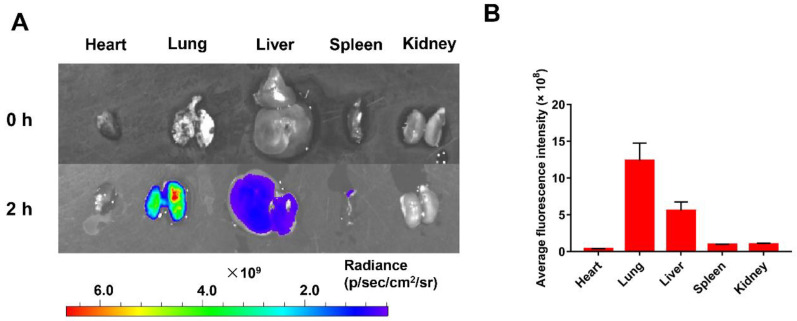
In Vivo distribution of nanoparticles. (**A**) Fluorescence intensity images of main organs of mice after intravenous injection of Cy5-linked TPT@CL-MOF at 2 h. (**B**) Average fluorescence intensity of main organs after 2 h of intravenous injection (*n* = 3).

**Figure 5 pharmaceutics-15-00142-f005:**
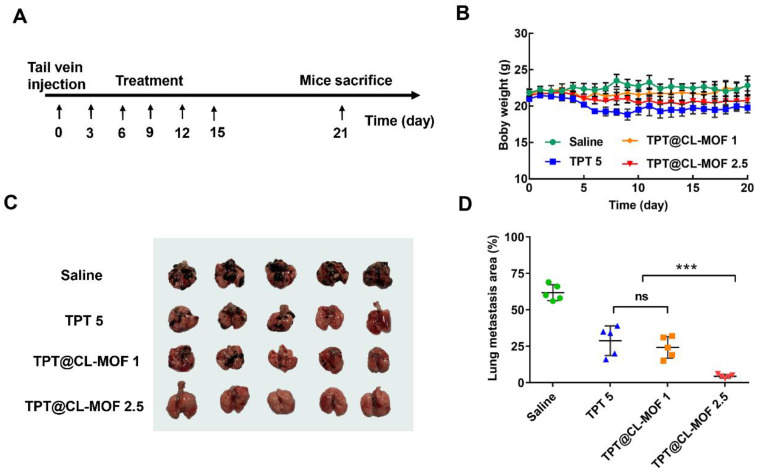
Therapeutic activity of TPT@CL-MOF against B16F10 melanoma lung metastases. (**A**) Diagram of dosing regimen. (**B**) Body weight of mice receiving different treatments. (**C**) Photographs of lungs isolated from mice bearing B16F10 lung tumor after different treatments (*n* = 5). (**D**) Lung metastasis foci area from mice bearing B16F10 lung tumor after different treatments. TPT@CL-MOF 1 and TPT@CL-MOF 2.5 denotes TPT at 1 mg/kg and 2.5 mg/kg, respectively. *** *p* < 0.001. The abbreviation “ns” indicates no significant difference between relevant treatment groups.

**Figure 6 pharmaceutics-15-00142-f006:**
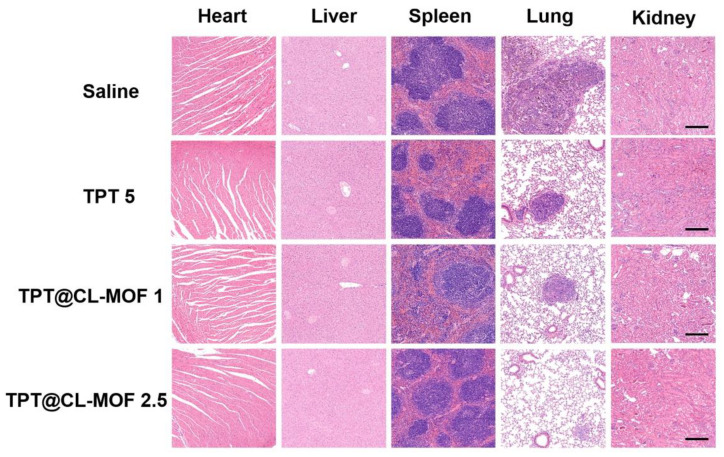
Histological observation of heart, liver, spleen, lung, and kidney. Scale bars, 200 µm. TPT@CL-MOF 1 and TPT@CL-MOF 2.5 denotes TPT at 1 mg/kg and 2.5 mg/kg, respectively.

## Data Availability

Data available on request.

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
