# Peer review of "Lactone Stabilized by Crosslinked Cyclodextrin Metal-Organic Frameworks to Improve Local Bioavailability of Topotecan in Lung Cancer"

_pharmaceutics, 2022, doi:10.3390/pharmaceutics15010142_

Round 1

Reviewer 1 Report

The manuscript Lactone stability enhanced by crosslinked cyclodextrin metal-organic frameworks to improve local bioavailability in lung cancer .I suggest accepting this manuscript after the revision, and the authors should consider the suggestions described below:
The paper needs to be improved in the following manners, this is a nice study, however, following questions are necessary to be answered before further processing

1.    Please briefly provide more about the main findings in the abstract.
2. Please use current and recent work in the literature review to give the reader a better idea.

1: Anti-bacterial/fungal and anti-cancer performance of green synthesized Ag nanoparticles using summer savory extract
2: Green Synthesis of Magnetic Nanoparticles Using Satureja hortensis Essential Oil toward Superior Antibacterial/Fungal and Anticancer Performance
3: Recent advancements in polythiophene-based materials and their biomedical, geno sensor and DNA detection
4: Antiproliferative and Apoptotic Effects of Graphene oxide@AlFu MOF based saponin natural product on OSCC Line
5: Bioactive Graphene Quantum Dots Based Polymer Composite for Biomedical Applications
6: Antibacterial effects of green-synthesized silver nanoparticles using Ferula asafoetida against Acinetobacter baumannii isolated from the hospital environment and assessment of their cytotoxicity on the human cell lines

3. The conclusions could be more specific and precise, I would suggest thinking about it.
4. Please go into more detail about the new aspect of your work at the end of the introduction.
5. Figures and tables could be checked once for resolution and correctness.
6. In the abstract, please provide some insight into the results of your work.
7. A detailed section on abbreviations should be updated.

Author Response

Reply to Reviewer 1:

Q1: Please briefly provide more about the main findings in the abstract.

A1: Thanks for your kind suggestion. “Especially, the formulation offered excellent protection in vitro against hydrolysis and increased TPT half-life from approximately 0.93 h to 22.05 h, which can be attributed to the host-guest interaction between cyclodextrin and TPT, as confirmed by molecular docking.” and “Moreover, the TPT@CL-MOF was efficiently distributed in the lungs after intravenous injection” have been added to the abstract (Page 2, Line 32-33 and Line 35-36 in revised manuscript).

Q2: Please use current and recent work in the literature review to give the reader a better idea.

1: Anti-bacterial/fungal and anti-cancer performance of green synthesized Ag nanoparticles using summer savory extract

2: Green Synthesis of Magnetic Nanoparticles Using Satureja hortensis Essential Oil toward Superior Antibacterial/Fungal and Anticancer Performance

3: Recent advancements in polythiophene-based materials and their biomedical, geno sensor and DNA detection

4: Antiproliferative and Apoptotic Effects of Graphene oxide@AlFu MOF based saponin natural product on OSCC Line

5: Bioactive Graphene Quantum Dots Based Polymer Composite for Biomedical Applications

6: Antibacterial effects of green-synthesized silver nanoparticles using Ferula asafoetida against Acinetobacter baumannii isolated from the hospital environment and assessment of their cytotoxicity on the human cell lines

A2: Thanks for your suggestion. The references concerning MOF and graphene quantum dots mentioned by you have been properly cited in the introduction of the revised manuscript. “Bioactive Graphene Quantum Dots Based Polymer Composite for Biomedical Applications” was cited in Line 61 (No.11) of the revised manuscript. “Antiproliferative and Apoptotic Effects of Graphene oxide@AlFu MOF based saponin natural product on OSCC Line” was cited in Line 63 (No.12) of the revised manuscript.

Q3: The conclusions could be more specific and precise, I would suggest thinking about it.

A3: Thank you for your valuable suggestion. The conclusions were improved in accordance with your suggestion (Page 19-20, Line 439-449 in revised manuscript).

    The improved conclusions are “In summary, this study highlighted the effectiveness of encapsulation of TPT within the porous structure of CL-MOF to enhance its local bioavailability for treating lung cancer. It was found that TPT@CL-MOF not only inhibited restrained burst drug leakage but also effectively improved the stability of TPT in physiological conditions by increasing the half-life from approximately 0.93 h to 22.05 h. In addition, TPT@CL-MOF exhibited excellent anticancer effects. TPT@CL-MOF significantly inhibited the migration and invasion of B16F10 cells in vitro and suppressed tumor growth with equivalent efficacy at a 5-fold reduced dose on a B16F10 pulmonary metastatic tumor model. Importantly, TPT@CL-MOF possessed excellent biocompatibility in recipient mice. Overall, TPT@CL-MOF might act as a unique and promising nanoplatform for pulmonary metastatic cancer treatment”.

Q4: Please go into more detail about the new aspect of your work at the end of the introduction.

A4: These details have also been added to the end of the introduction.

   “In this study, we developed an alternative pulmonary delivery platform that uses CL-MOF technology to capture and stabilize the physiologically unstable TPT in order to increase its local bioavailability. Subsequently, the TPT-loaded crosslinked cyclodextrin metal-organic framework (TPT@CL-MOF) was systematically characterized, and the release profile and stability tests were assessed under physiological conditions. As a proof-of-concept, we investigated the in vitro inhibitory effects assay of TPT@CL-MOF on murine melanoma cells (B16F10), as well as the therapeutic efficacy in the B16F10 metastatic tumor-bearing mice.” was add in page 5 lines 90-97 in revised manuscript.

Q5: Figures and tables could be checked once for resolution and correctness.

A5: Thanks for your valuable suggestion. Figures have been modified to improve the resolution and correctness.

Q6: In the abstract, please provide some insight into the results of your work.

A6: Thank you for your valuable suggestion. “Therefore, this study provides a possible alternative therapy for the treatment of lung cancer with the camptothecin family drugs or other unstable therapeutically significant molecules.” has been added to the abstract based on the results (Page 2, Line 39-41 in revised manuscript).

Q7: A detailed section on abbreviations should be updated.

A7: We appreciate for your indication. All abbreviations were updated.

Reviewer 2 Report

Comments to the Authors:

The authors presented study on design of crosslinked cyclodextrin metal-organic frameworks for improvement of drug (Topotecan) local bioavailability in lung cancer. The topic is interesting regarding the potential use of the investigated metal-organic frameworks as prospective drug carriers. The topic of the manuscript is within the scope of Pharmaceutics journal.

However, there are some uncertainties in several parts of the manuscript and certain points need to be clarified, according to the specific comments below.

1.      The change of the Title of the manuscript is recommended, so that it would be in accordance with the presented investigation. Namely, it would be much more representative to point out that the drug topotecan was used as an active pharmaceutical ingredient.

2.      In the Materials and methods section, it is necessary to describe in detail the procedure of CL-MOF synthesis. Namely, it is necessary to review and correct the following sentences:

Briefly, 3.1 g of CD-MOF and 3.0 g DPC at a 1:6 molar ratio was incubated in 40 mL of DMF under the string” - it is unclear what is string?

„Then, the mixture was able to proceed for 24 h under stirring at 80 °C.“ - it is necessary to explain how it was ensured that the solvent does not evaporate?

“Herein, 30 mg TPT was dissolved into 3 mL distilled water containing 50 mg CL-

MOF.” - the sentence is not clearly written, please correct it.

„The resulting dispersion was incubated for 12 h at ambient temperature in the dark. The TPT-loaded nanoparticles (TPT@CL-MOF) were washed to remove the unloaded TPT and obtained by centrifugation and lyophilization.“ - the conditions under which the incubation was carried out should be clearly stated. In the second sentence, it is not clearly when centrifugation and lyophilization were performed.

Finally, it is not clear from Equation (1), how the TPT loading was determined from the calibration curve, considering that the masses are given in the equation.

In the section 2.6 In vitro release profiles it is necessary to describe the HPLC method used for the determination of TPT.

In the section 2.9 Cytotoxicity assay, it is necessary to state the range of TPT@CL-MOF concentrations used in the assay.

In the section 2.12 Anti-metastatic assays, it is stated that: “Three days later, the mice were randomly divided into five groups for treatment with saline, free TPT (5 mg/kg), TPT@CL-MOF (with TPT at 2.5 or 1 mg/kg), and administration frequency was once every three days for five sequential cycles.” - it is necessary to describe in which from free TPT (5 mg/kg) and TPT@CL-MOF (with TPT at 2.5 or 1 mg/kg) were administered?

3.      In the Results and discussion section it is necessary to explain why a TPT loading content of 12.3% is considered as a satisfactory.

In the part where the results of the DSC analysis are interpreted, it is stated that The endothermic peak of TPT in TPT@CL-MOF completely disappeared, indicating an apparent change of the original crystal into amorphous form when encapsulated into the CL-MOF matrix.“ – according to literature data, can TPT exist in both crystalline and amorphous form? Is there only one amorphous form?

In Figure S2, please consider presenting the data on x-axis as square root of time (for Higuchi model) and log time (for Ritger-Peppas model).

In the section:

„Hydrolysis of free TPT generated rapidly at pH 7.4 with a short t1/2 of about 0.93 h. Fortunately, more lactone fraction of TPT was supervised from TPT@CL-MOF than free TPT at every designated time point in the PBS of pH 7.4, which possessed a long t1/2 of 22.05 h. It was attributed to the pore canal of CL-MOF avoiding the direct contact of TPT molecules with the external aqueous interface. But there was still a portion of TPT existed in the opening-ring carboxylate form. The reason may be that these released TPT contacted with the external buffer, even if they were initially encapsulated in the cavity.“ – please add, standard deviations at log t1/2 values. Furthermore, it should be considered in more detail, the statement related to contact of the external buffer with the opening-ring carboxylate form of TPT.

In several places in the Results and discussion, TPT-formulations are mentioned. It is necessary to correctly state on which of the examined samples this refers to.

In the captions of Figures 5 and 6, an explanation for the labels TPT@CL-MOF1 and TPT@CL-MOF 2.5 should be added.

Figure S4 is not mentioned in the manuscript. Additionally, the presented IC50 of TPT and TPT@CL-MOF to B16F10 cells, are slightly different. Please, add and explanation.

4.      English should be improved all over the text and checking for spelling and grammar is recommended.

Round 2

Reviewer 2 Report

The Authors have answered most of the questions raised after the previous revision. However, there are still uncertainties in several parts of the manuscript. Therefore, please make the necessary changes.
